# Cavity-coupled telecom atomic source in silicon

Adam Johnston [1,2,4], Ulises Felix-Rendon [1,2,4], Yu-En Wong [1,2,4] & Songtao Chen [1,3] ✉

Novel T centers in silicon hold great promise for quantum networking applications due to their telecom band optical transitions and the long-lived ground state electronic spins. An open challenge for advancing the T center platform is to enhance its weak and slow zero phonon line (ZPL) emission. In this work, by integrating single T centers with a low-loss, small mode-volume silicon photonic crystal cavity, we demonstrate an enhancement of the fluorescence decay rate by a factor of $F = 6.89$. Efficient photon extraction enables the system to achieve an average ZPL photon outcoupling rate of 73.3 kHz under saturation, which is about two orders of magnitude larger than the previously reported value. The dynamics of the coupled system is well modeled by solving the Lindblad master equation. These results represent a significant step towards building efficient T center spin-photon interfaces for quantum information processing and networking applications.

Optically interfaced atomic defects in solid-state materials are important building blocks for a variety of quantum technologies[1]. For example, nitrogen and silicon vacancy centers in diamonds have been used to demonstrate milestone results for fiber-based quantum networks, including spin-photon entanglement[2], deterministic entanglement generation between remote spins[3], quantum state teleportation[4], and memory-enhanced communication[5]. However, these defects have optical transitions at the visible or near-infrared spectral range resulting in large fiber transmission loss, requiring nonlinear frequency conversion[6] to extend the network range. Significant progress has been made towards utilizing atomic defects with telecom optical transitions, leading to the discovery of single vanadium ions in silicon carbide[7], defects in gallium nitride[8], and single erbium ions in yttrium orthosilicate[9], as well as a recent demonstration of indistinguishable telecom photon generation from single erbium ions[10].

Early exploration into the scalable fabrication of atomic defects with photonic structures relies on heterogeneous material integration and typically involves pick-and-place type of fabrication procedures[9,11,12]. On the other hand, silicon-on-insulator (SOI) is a mature and scalable platform to enable large-scale monolithic

photonic and electronic device integration. Telecom-interfaced solid-state spins in silicon can thus benefit from the technological advantages of the SOI platform, and be utilized for realizing large-scale spin-based integrated quantum photonic chips[13]. Moreover, silicon can be isotopically enriched to create a "semiconductor vacuum"[14] for lowering the magnetic noise generated from the $^{29}$Si nuclear spin bath.

Beyond efforts towards optical addressing of erbium ions in silicon[15–17], multiple novel atomic defect centers in silicon, including C, G, T, and W centers[18–27], have been experimentally identified recently towards quantum information applications. Among them, T centers are particularly promising due to their telecom O-band optical transitions, doublet ground state spin manifold, and long spin coherence times in an enriched $^{28}$Si sample[23]. Single T centers have been isolated in micropuck[24] and waveguide[28] structures. To further advance the single T center platform for quantum networking applications, challenges remain to enhance its weak and slow coherent emission at the zero phonon line (ZPL). The cavity-induced Purcell effect[29] has been widely used for enhancing the fluorescence emission of various atomic defects in solids, including G centers[30–32].

In this work, we demonstrate Purcell enhancement of a single T center in a low-loss, small mode-volume silicon photonic crystal (PC)

[1]Department of Electrical and Computer Engineering, Rice University, Houston, TX 77005, USA. [2]Applied Physics Graduate Program, Smalley-Curl Institute, Rice University, Houston, TX 77005, USA. [3]Smalley-Curl Institute, Rice University, Houston, TX 77005, USA. [4]These authors contributed equally: Adam Johnston, Ulises Felix-Rendon, Yu-En Wong. ✉e-mail: songtao.chen@rice.edu

cavity. When the cavity is tuned into resonance with the single T center we observe an enhancement of its fluorescence decay rate by a factor of $F = 6.89$, shortening the single T center lifetime to $136.4 \pm 0.6$ ns. Leveraging the nanophotonic circuit and an angle-polished fiber for light coupling[33], the system detection efficiency reaches $\eta_{sys} = 9.1\%$, representing the probability of a single photon emitted into the cavity being registered by the detector. This efficiency is 20-fold larger than that achieved in a typical confocal-type measurement system for T centers. We probe single T centers in the device using time-resolved photoluminescence excitation (PLE) spectroscopy. Under the pulsed excitation, the system can detect 0.01 ZPL photon per excitation, reaching an average photon count rate of 73.3 kHz, which is about two orders of magnitude improvement from the previously reported emission rate for single T centers in the waveguide[28]. By solving the Lindblad master equation, we develop a numerical model to describe the coupling dynamics between the single T center and the cavity, and extract cavity-QED parameters as well as T center pure dephasing rate ($\Gamma_d$) and spectral diffusion ($\Gamma_{sd}$). This work represents a key step towards utilizing single T centers in silicon for quantum information applications.

## Results

### Device integration and PLE spectroscopy

Our experimental configuration is outlined in Fig. 1a. The nanophotonic devices are fabricated on a SOI sample which is situated on the cold finger of a closed-cycle cryostat ($T = 3.4$ K). Each device consists of a subwavelength grating coupler (GC)[34] and a one-dimensional PC cavity, which is connected by a linearly tapered waveguide (Fig. 1b). Optical coupling to the PC cavities is accomplished by using an angle-polished fiber via the GC with a one-way coupling efficiency of $\eta_{GC} = 46.1\%$ at 1326 nm (Fig. 1c). The fiber is mounted on a three-axis translation stage for optimizing the coupling. The PC cavity used in this work (Fig. 1d) has a quality factor $Q = 4.3 \times 10^4$. Fluorescence from the T center is detected by a fiber-coupled superconducting nanowire single photon detector (SNSPD) located in a separate cryostat (Fig. 2a). To match the atomic transition, we coarsely red-tune the cavity resonance by condensing nitrogen gas onto the surface of the device; we fine blue-tune the cavity resonance by sending laser pulses with a high optical power into the cavity (Supplementary Section 1.2).

T centers are generated in the middle of the device layer (220 nm in thickness) of the SOI samples using the ion implantation method[35].

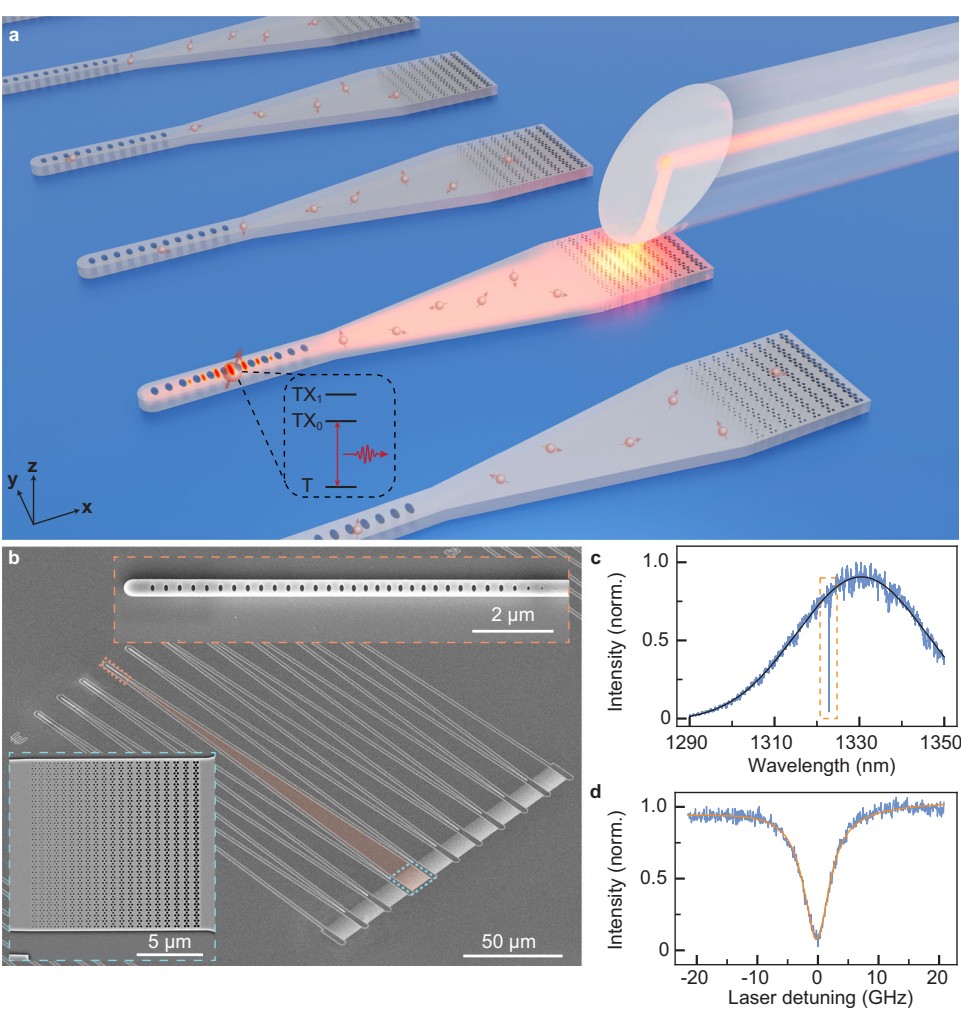

**Fig. 1 | Efficient optical coupling to single T centers in a silicon photonic cavity. a** Schematic illustration of fiber coupling to the nanophotonic circuits. Each PC cavitiy and subwavelength GC (with a width of 13.1 μm) are connected via a linearly tapered waveguide (200 μm in length). The GC allows efficient light coupling with an angle-polished fiber. $(x, y, z)$ refer to ($[0\bar{1}0]$, $[100]$, $[001]$) directions of the silicon crystal. T centers (solid red balls with arrows) are uniformly generated across the whole SOI device layer using the ion implantation method. Inside the dashed square, the simplified electronic level structure of the single T center is shown[23]. **b** Scanning electron microscope image of a block of 10 devices. An individual device (orange-shaded) consists of a PC cavity on the upper left end (inside the orange dashed rectangle) and a GC on the lower right end (inside the blue dashed rectangle). **c** Measured GC coupling spectrum (blue), with a Gaussian (black) fitted FWHM linewidth of 36.1 nm. The orange dashed box encloses a PC cavity. **d** Reflection spectrum of the cavity (blue) shown in panel (**c**), with a Lorentzian (orange) fitted FWHM linewidth of $5.22 \pm 0.04$ GHz. This cavity is used in later experiments. Results shown in panel (**c**) and (**d**) are measured at $T = 3.4$ K.

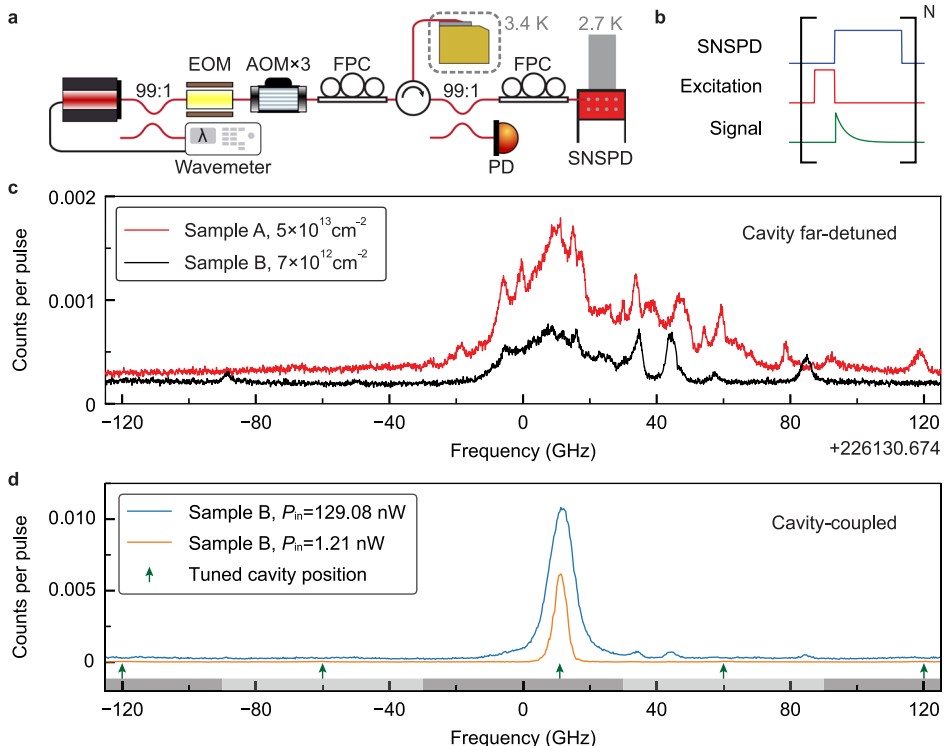

**Fig. 2 | Photoluminescence excitation spectroscopy for T centers. a** Simplified schematic of the experimental setup for performing PLE spectroscopy. Short excitation pulses are generated via a series of acousto-optic modulators (AOMs) and an electro-optic modulator (EOM) from a stabilized laser. The EOM is also used to generate laser sidebands. Fluorescence photons are redirected via an optical circulator to a SNSPD in another cryostat with $T = 2.7$ K for detection. A small portion of the signal enters a photodiode (PD) for monitoring cavity reflection. FPC: fiber polarization controller. **b** PLE pulse sequence. The SNSPD is gated to prevent detector latching due to the strong laser excitation pulses. **c** PLE spectrum for two different implanted SOI samples under similar excitation conditions when the

cavity is far-detuned ($\Delta_{ac} > 200$ GHz). The vertical axis indicates the average number of photons detected in an 5.55 μs integration window after each excitation pulse. Both samples are implanted with 1:1 ratio of $^{12}$C and $^{1}$H, and the implantation fluences are labeled in the legend. All the frequencies in this paper, if not specified, are offset from a reference $f_0 = 226130.674$ GHz (1325.749 nm or 935.201 meV). **d** PLE spectrum for sample B at different excitation powers when the cavity is tuned within the scanning range. Each tuned cavity position is indicated by a green arrow, with the corresponding laser scanning range shaded on the $x$-axis under the arrow. The spectrum shows a dominant peak, which we interpret as a single T center coupled to the cavity.

Each SOI sample is implanted sequentially with 1:1 ratio of 35 keV $^{12}$C and 8 keV $^{1}$H ions, with rapid thermal annealing performed in between and after the implantation steps (Supplementary Section 3.1). The two samples shown in this work have an implantation fluence of $5 \times 10^{13}$ cm$^{-2}$ (sample A) and $7 \times 10^{12}$ cm$^{-2}$ (sample B), respectively, which result in different T center densities after the generation process. To search for T centers, we perform time-resolved PLE spectroscopy by scanning the wavemeter-stabilized laser around the reported inhomogeneous center of T centers in silicon[23,24,35], with the pulse sequence shown in Fig. 2b.

First, we measure the spectrum with the cavity far-detuned from the scan range (Fig. 2c). The inhomogeneous distribution linewidth is about $\Gamma_{inh} \approx 29$ GHz. Isolated peaks can be observed away from the inhomogeneous center, which we interpret as the optical transitions of T centers that are likely located inside the taper waveguide. These peaks have an average full-width half maximum (FWHM) linewidth of $2.40 \pm 0.86$ GHz, and a fluorescence lifetime of $836.8 \pm 57.3$ ns (Supplementary Section 4), which is slightly shorter than the bulk T centers' lifetime of 940 ns[23]. The estimated T center densities are $\sim 1 \times 10^{12}$ cm$^{-3}$ and $\sim 3 \times 10^{11}$ cm$^{-3}$ for sample A and B, respectively (Supplementary Section 3.1). To gauge the probability of an excited T center emitting a photon into the waveguide mode, we analyze a typical waveguide-coupled T center (at 46 GHz in sample B); we use the bounded T center quantum efficiency (discussed below) to estimate its emission efficiency to the waveguide mode as $2.6\% \leq \eta_{wg} \leq 10.9\%$ (Supplementary Section 4).

Next, we scan the laser frequency with the cavity tuned in-range to obtain the cavity-coupled PLE spectrum (Fig. 2d). In sample B, a new T center peak at the inhomogeneous center emerges with its fluorescence significantly surpassing all other peaks. This cavity-coupled T center has a FWHM linewidth of $\Gamma = 3.81 \pm 0.07$ GHz under a low excitation power (Fig. 3a). To verify the peak originates from a single T center, we measure the second-order autocorrelation function $g^{(2)}$ using all the detected fluorescence photons after each excitation pulse (Fig. 3b). Photon antibunching is observed with the value $g^{(2)}(0) = 0.024 \pm 0.018$, which confirms the majority of the detected photons come from a single emitter. This is the lowest $g^{(2)}(0)$ value ever observed for single T centers, and is comparable or better than other defect-based telecom emitters in solids[7–10,20,27,36]. Autocorrelation measurements for the single T center can show bunching ($g^{(2)}(n) > 1$, when $|n| \geq 1$) under certain excitation conditions, which we speculate to be caused by spectral diffusion (Supplementary Section 5.3).

The emission amplitude of the cavity-coupled single T center saturates at 0.01 photons per excitation pulse (Fig. 3c). Both the saturation and power-dependent linewidth (Fig. 3d) are well described by the numerical modeling (discussed below). The measured $g^{(2)}(0)$ at higher powers (Fig. 3d) is limited by the accidental coincidences from background T centers' emission (Supplementary Section 5.2). To characterize the spectral diffusion, we monitor the spectrum of the cavity-coupled single T center over a few hours time span by taking repetitive PLE scans (Fig. 3e), which reveals a spectrum-center distribution of $11.30 \pm 0.13$ GHz. A similar level of spectral diffusion is

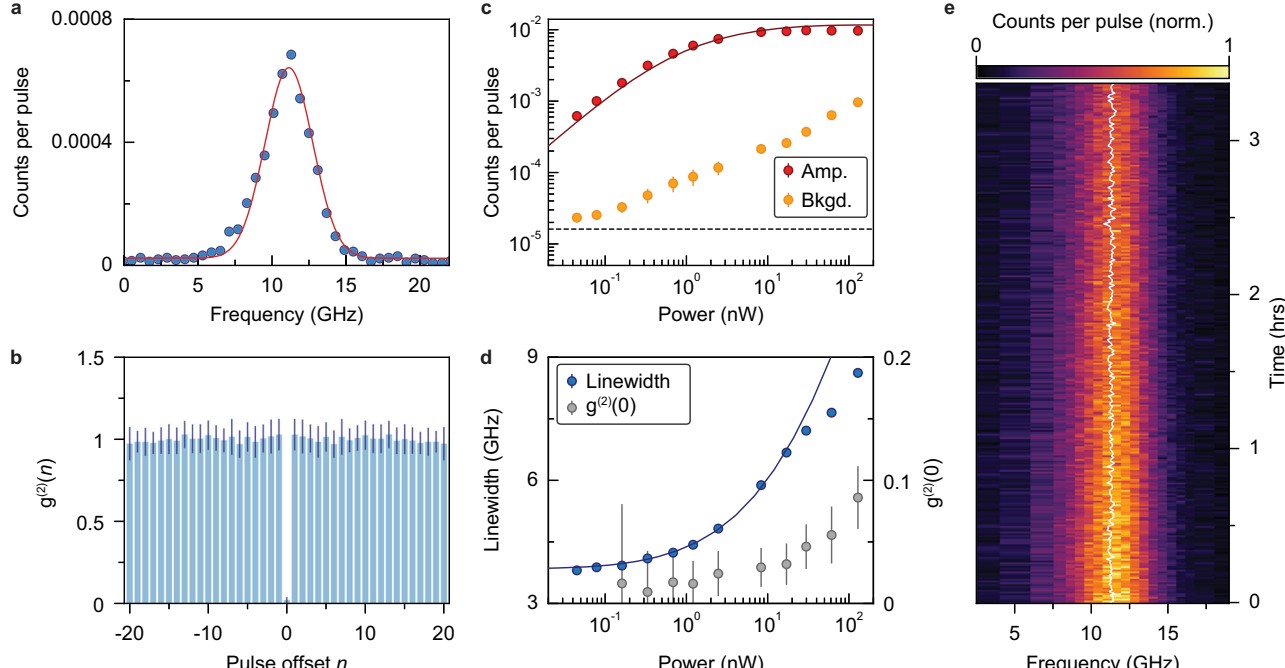

**Fig. 3 | Cavity-coupled single T center characterizations. a** PLE spectrum of the cavity-coupled single T center under an excitation power $P_{in} = 0.04$ nW. The red line shows the Gaussian fitting. **b** Second-order autocorrelation measurement for the same T center ($P_{in} = 2.47$ nW) shows a strong antibunching. All the photons detected after each excitation pulse are binned into a single-time bin. The horizontal axis shows the autocorrelation offset in units of the pulse repetition period (8 μs). **c** Gaussian-fitted amplitude (Amp.) and background (Bkgd.) of single T-center PLE spectrum at different $P_{in}$. The background has the main contribution from other weakly-coupled T centers beyond the SNSPD dark counts (3 Hz, black dashed line). The saturation behavior is well described by the numerical model (red line). **d** Spectrum linewidth of the single T center and the $g^2(0)$ at different $P_{in}$. The blue line shows the numerical calculation results. For results shown in panel (**a**–**d**), the cavity is tuned into resonance with the T center transition ($\Delta_{ac} = 0$). **e** Spectral diffusion of the single T center ($P_{in} = 8.31$ nW). White line shows the center of the spectrum at each iteration (with a duration of 42 seconds) of the experiment. In all plots, error bars denote $\pm 1\sigma$ statistical uncertainty.

observed for waveguide-coupled T centers (Supplementary Section 4). We note that this method only provides a lower bound of the $\Gamma_{sd}$ due to the limited experiment repetition rate. We later turn to the numerical modeling to extract $\Gamma_{sd}$.

We apply a magnetic field ($B$) up to 300 mT along silicon [100] direction aiming to split the single T center line. We note that we have not been able to observe unambiguous Zeeman splitting using simultaneous two-tone laser sideband excitation, which is likely due to the limited splitting compared with the broad single T center linewidth. When using the single-tone laser excitation, the PLE amplitude decreases at increasing $B$ field due to spin polarization[35]. We model this behavior (Supplementary Section 5.4) to extract the difference of the excited- and ground-state g-factors $|\Delta_g| = 0.55 \pm 0.04$, which matches with one of the two predicted $|\Delta_g|$ values for T centers under a $B$ field along the silicon [100] direction[35].

## Purcell enhancement and numerical modeling

Lastly, we study the cavity-QED of the coupled system. When the cavity is tuned into resonance, the single T center's fluorescence lifetime is shortened to $136.4 \pm 0.6$ ns (Fig. 4a), which is $6.89 \pm 0.03$ times faster than the bulk lifetime of $1/\Gamma_0 = 940$ ns[23]. Leveraging this enhanced decay, we extract an average ZPL photon outcoupling rate of 73.3 kHz at saturation for the cavity-coupled single T center. To confirm the enhancement originates from the cavity coupling, we measure the fluorescence decay rate $\Gamma_{cav}$ at different cavity detunings ($\Delta_{ac}$) (Fig. 4b), which can be described as $\Gamma_{cav}/\Gamma_0 = P_t/[1 + (2\Delta_{ac}/\tilde{\kappa})^2] + \Gamma_\infty/\Gamma_0$, where $P_t = 5.88 \pm 0.04$ is the Purcell factor describing the fluorescence decay enhancement due to the cavity, $\Gamma_\infty = (1.03 \pm 0.02)\Gamma_0 \approx \Gamma_0$ is the asymptotic decay rate at large detunings, and $\tilde{\kappa}/2\pi = 7.11 \pm 0.09$ GHz is the characteristic linewidth. To explain the deviation of $\tilde{\kappa}$ from the cavity linewidth $\kappa/2\pi = 5.22$ GHz, we turn to numerical calculations by solving the Lindblad master equation. Beyond the cavity and atomic loss channels, we also incorporate the pure dephasing and spectral diffusion processes (Supplementary Section 6.1). The dynamics of the coupled system can be described by the Jaynes-Cummings Hamiltonian of the form,

$$H/\hbar = \Delta_a \sigma_+ \sigma_- + \Delta_c a^\dagger a + g(\sigma_+ a + \sigma_- a^\dagger) + \frac{\Omega}{2}(\sigma_+ + \sigma_-), \quad (1)$$

which assumes rotating wave approximation in the rotating frame of the laser field ($\omega_L$). Here $\Delta_a = \omega_a - \omega_L$ and $\Delta_c = \omega_c - \omega_L$ are, respectively, the detunings of the laser from the T center transition $\omega_a$ and the cavity resonance $\omega_c$; $g$ is the coupling rate between the single T center and the cavity mode, and $\Omega$ is the optical Rabi frequency. The global fitting of the experimental data based on the numerical calculations given the known $\kappa$ and $\Gamma_0$ (Supplementary Section 6.1), reveals the full cavity-QED parameter set $(g, \kappa, \Gamma_0) = 2\pi \times (42.4$ MHz, 5.22 GHz, 169.3 kHz), an excited-state dephasing rate $2\Gamma_d = 2\pi \times 1.29$ GHz, and a spectral diffusion $\Gamma_{sd} = 2\pi \times 1.69$ GHz. The characteristic linewidth $\tilde{\kappa}$ has contributions from the cavity linewidth $\kappa$ as well as $\Gamma_d$ and $\Gamma_{sd}$. The model can simultaneously capture the saturation (Fig. 3c), power-dependent linewidth (Fig. 3d), and detuning-dependent fluorescence decay (Fig. 4b) results. Beyond the dephasing and the spectral diffusion, the single T center linewidth also has a minor contribution from the thermal broadening, which we estimate as $\Gamma_{th} \sim 0.1$ GHz from the temperature-dependent linewidth measurements (Supplementary Section 5.5).

To find out the Purcell enhancement of the single T center ZPL ($P_{ZPL}$) and its quantum efficiency ($\eta_{QE}$), we express the total emission rate of a single T center in absence of a cavity as the summation of the emission rates into the ZPL and the phonon sideband (PSB), as well as

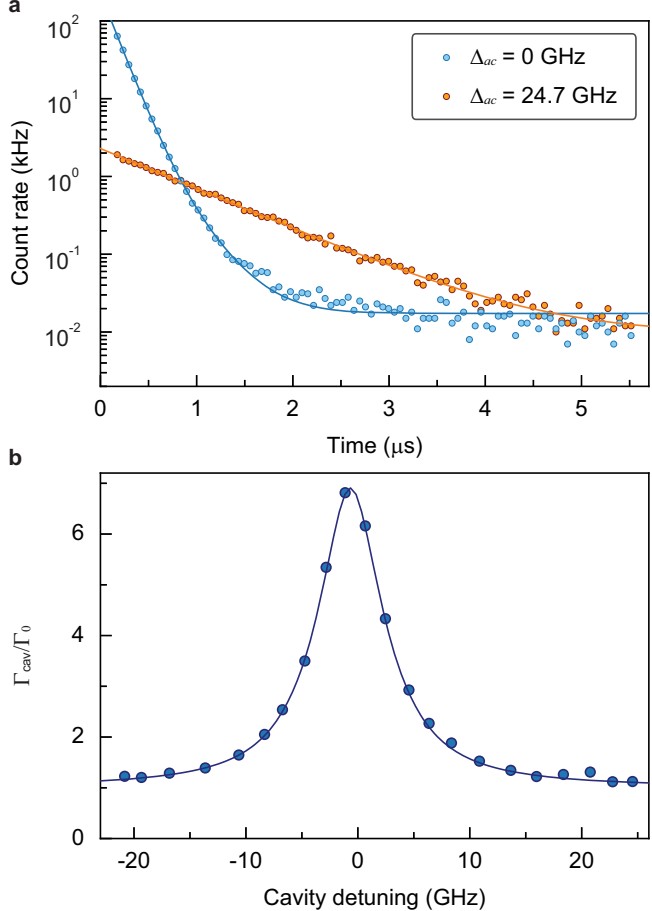

**Fig. 4 | Cavity-enhanced fluorescence emission of the single T center. a** Time-resolved fluorescence for the cavity-coupled single T center under saturation ($P_{in}$ = 17.01 nW) with different cavity detunings. The exponential fitting (blue and orange lines) reveals decay lifetimes of 136.4 ± 0.6 ns and 835.2 ± 3.1 ns for $\Delta_{ac}$ = 0 GHz and $\Delta_{ac}$ = 24.7 GHz, respectively. **b** Decay rate enhancement at different cavity detunings with the laser fixed at the T center transition under an excitation power $P_{in}$ = 1.21 nW. To gain better tuning accuracy, the cavity resonance is first red-tuned to $\Delta_{ac}$ = −20.8 GHz, and subsequently blue-tuned. The blue line shows the numerical calculation results based on solving the Lindblad master equation.

nonradiative relaxation[37], $\Gamma_0 = \gamma_{ZPL} + \gamma_{PSB} + \gamma_{nr}$. The Debye-Waller (DW) factor and the quantum efficiency can then be defined as DW = $\gamma_{ZPL}/(\gamma_{ZPL} + \gamma_{PSB})$ and $\eta_{QE} = (\gamma_{ZPL} + \gamma_{PSB})/\Gamma_0$, respectively. When the cavity is tuned into resonance with the T center ZPL, the cavity-enhanced decay rate is $\Gamma_{cav} = (P_{ZPL} + 1)\gamma_{ZPL} + \gamma_{PSB} + \gamma_{nr}$, where $P_{ZPL} = P_t/(DW\eta_{QE})$ is the Purcell factor describing the enhancement of the ZPL. We can thus put a lower bound on the $P_{ZPL} \geq P_t/DW = 25.6$, using the reported DW of 23%[23]. For simplicity, we neglect the potential suppression of the $\gamma_{PSB}$ due to the cavity[37]. The ratio of the single T center ZPL emission coupled to the resonant cavity mode can be estimated as $\beta = P_{ZPL}\gamma_{ZPL}/[(P_{ZPL} + 1)\gamma_{ZPL} + \gamma_{PSB} + \gamma_{nr}] = P_t/(P_t + 1) = 85.5\%$. Due to sub-optimal positioning of the single T center inside the cavity and imperfect dipole alignment with the local cavity electrical field polarization, the $P_{ZPL}$ extracted from measurements should be smaller than the simulated Purcell ($P_{ZPL} \leq P_{ZPL}^{sim}$). This enables us to put a lower bound on the quantum efficiency $\eta_{QE} \geq 23.4\%$ for the single T center (Supplementary Section 2.1).

## Discussion

We now discuss pathways to improve the performance of the cavity-coupled single T center system. We note that the linewidth of the observed single T center is significantly larger than the

Purcell enhanced lifetime-limited linewidth ($2\pi \times 1.2$ MHz). One culprit is the fast dephasing process, which necessitates further investigation to reveal its origin. Significant reduction of $\Gamma_{inh}$ down to 33 MHz has been demonstrated for ensemble T centers in enriched $^{28}$Si[23]. In future work, SOI samples with an enriched silicon device layer can be prepared via molecular beam epitaxy[38] to minimize the dephasing. Furthermore, local electrodes can be fabricated on the SOI device layer for implementing electrical field control to minimize the spectral diffusion via in situ tuning[39] or depletion of the charge noises[40]. Lastly, focused-ion-beam-based[41] and masked[42] ion implantation can be leveraged to increase the yield of T center generation at the cavity center.

In summary, we have demonstrated enhanced light-matter interaction for a single T center by integrating it with a silicon nanophotonic cavity. This work opens the door to utilize single T centers in silicon for quantum information processing and networking applications. With realistic improvements in the quality factor of the optical cavity ($Q = 5 \times 10^5$) and narrower linewidth in an enriched sample ($\Gamma \sim 10$ MHz), a large atom-cavity cooperativity $C \geq 29$ can be expected, which can lead to applications for high-fidelity dispersive spin readout[43] and cavity-mediated spin-spin interactions[44]. Moreover, the current approach can enable parallel control and readout of multiple T centers in the cavity via the frequency domain addressing technique[45]. Finally, leveraging the mature silicon photonics technology, small-footprint and scalable T-center-spin-based silicon quantum photonic chips[13] may be envisioned.

*Note*: While finalizing this manuscript, we became aware of a related publication on the detection of a single T center coupled to a cavity using above-band excitation[46].

## Methods

### Device nanofabrication

All of the nanophotonic devices are fabricated on SOI samples (WaferPro). The SOI has a 220 ± 10 nm float zone grown P-type device layer with resistivity ≥1000 $\Omega \cdot$ cm. The buried oxide has a customized thickness of 2.3 μm for maximizing the GC coupling efficiency, and the handling layer has a thickness of 725 μm. We spin coat 400 nm electron beam (ebeam) resist ZEP520A (Zeon Specialty Materials Inc.) onto 9 × 9 mm² SOI chips and bake at 170 °C for 5 mins. The sample is exposed using an Elionix ELS-G100 ebeam writer with a dosage of 225 μC/cm², and subsequently developed in o-xylene at room temperature for 90 seconds and rinsed in isopropanol for 20 seconds. The pattern is then defined on the resist layer, which acts as the etching mask. The sample etching is performed using an inductively coupled plasma (ICP) reactive ion etcher (Oxford Plasmalab System 100/ICP 180) with $SF_6/C_4F_8$ gases. The sample is kept at 0 °C during the etching process. After etching, the sample goes through a series of processes including oxygen plasma descum, dicing (into 4.5 × 4.5 mm²), resist stripping, and piranha cleaning before being transferred into the cryostat for measurements.

### T center generation via ion implantation

T centers shown in this work are generated via a uniform ion implantation method following a published procedure[35]. We use an equal fluence for $^{12}$C and $^1$H during the ion implantation processes (II–VI Coherent Corp). We perform $^{12}$C ion implantation at 7° direction, 35 keV energy, followed by rapid thermal annealing (RTA) at 1000 °C for 20 s under an Ar background to repair lattice damage and substitute the implanted carbon[24]. Next, a second round implantation of $^1$H at 7° direction, 8 keV energy is performed. After the two implantation steps, we boil the sample for 1 h in DI water, followed by a second RTA process at 420 °C for 3 min with a $N_2$ background.

## Data availability

The data that support the findings of this study are openly available on the Harvard Dataverse at https://doi.org/10.7910/DVN/XCS15A.

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

## Acknowledgements

We gratefully acknowledge Han Pu, Alexey Belyanin, and Tanguy Terlier for helpful discussions, and John Bartholomew for feedback on a manuscript draft. Support for this research was provided by the National Science Foundation (NSF, CAREER Award No. 2238298), the Robert A. Welch Foundation (Grant No. C-2134) and the Rice Faculty Initiative

Fund. We acknowledge the use of cleanroom facilities supported by the Shared Equipment Authority at Rice University.

## Author contributions

A.J., U.F., Y.W. and S.C. contributed to the design and execution of the experiment. Y.W. and S.C. performed numerical modeling of the cavity-QED. A.J., U.F., Y.W. and S.C. analyzed the data and wrote the manuscript. S.C. supervised the whole project.

## Competing interests

The authors declare no competing interests.
