## [Peer Review File · Nature Communications]

REVIEWER COMMENTS

Reviewer #1 (Remarks to the Author):

Review: Cavity-coupled telecom atomic source in silicon

The article presents the embedment of Silicon T-centers in a photonic structure consisting of a nanobeam cavity, connected to a tapered waveguide and a grating to outcouple the photons into an optical fiber. On top of an impressive cleanroom work to fabricate the structure and generate the emitters, a methodical optical analysis has been performed, with a thorough estimation of the efficiency of the system, as well as complete characterization of one emitter. An impressive Purcell factor of 6 has been achieved, and all results have been compared to numerical modelling using JC Hamiltonian and Lindblad master equations.

The article is well structured, focused first on the fabrication and basic characterization of the sources, then on the estimation of the Purcell factor based on these measurements and on numerical modelling. While the article describes interesting results, it would further improve if the authors could clarify a certain number of points, see below:

The sample A presented in Figure 2c. does not seem to be used anymore afterward in the manuscript. If the only purpose of the sample A is to show that a higher fluence leads to a higher density of emitter, it would be helpful to mention it explicitly.

Comparing these densities and the dimensions of the cavity, it could have been interesting to hint the reader on a "fabrication yield", that is, a probability to generate an emitter at the correct location (and maybe polarization) inside the photonic crystal.

Right after mentioning the densities, you suddenly discuss the emission efficiency of a single emitter into the waveguide mode? The link with the beginning of the paragraph is not straightforward. Also, you evaluate this η_{wg} for a single T-center at 46 GHz using the quantum efficiency of another T-center (the one that you use for the rest of the article at 11.3 GHz), although it seems that the first one is located inside the taper while the second one is located inside the nanobeam PC? Would the η_{wg} of these two emitters be identical despite their different position and eventually polarization?

Indicating the detected count rates in pair with the counts per pulse could have been helpful for the readers, especially for such telecom applications where the actual count rate is of key importance.

Is the g_2 mentioned in the abstract obtained for the count rate mentioned right after?

Moreover, it could have been interesting to compare these g_2 and count rates not only to the T-center literature, but also to other deterministic telecom emitters from other materials (for instance based on GaN color centers: *sciadv.aar3580* and *Nanophotonics 12.8 (2023): 1405-1419.*)

In the JC Hamiltonian, it is not clear to me why the drive Hamiltonian is defined as function of the operators for the T-center 2-level system rather than of the cavity.

In figure 4a, the blue fitting curve (no detuning) does not seem to catch an apparent longer decay time (time delays above 1.5 ns). Is this second decay rate real or only related to measurement uncertainty. If this decay is related only to the wg-coupled T-centers, and considering the large repetition rate that you have, could not you integrate longer to get rid of that visible second decay? How long did you integrate for these measurements?

Despite these few remarks, the article is very complete, the writing is clear, the structure is logic, and each characterization is conducted rigorously.

Reviewer #2 (Remarks to the Author):

The authors have provided a timely, interesting, and thorough study of Purcell-enhanced T centres integrated into photonic devices. Their paper provides impressive device performance for an early demonstration, a careful analysis of their measurements, and a very helpful and appreciated SuppMat section. I certainly recommend this excellent work for publication in Nature Communications.

Although none of these questions should be considered blockers to publication, I do have the following questions for the authors:

- Line 119 of the main text: As written, it appears that the values of κ and Γ_0 were extracted from the fitting procedure, but these parameters were defined to have these values earlier in the paragraph. Were $\kappa/2\pi=5.22\text{GHz}$ and $\Gamma_0/2\pi=169.3\text{kHz}$ obtained from the global fit? Or were they fixed parameters in that procedure?
- On a related note, the bulk lifetime of 940ns used by the authors was measured with above-band excitation in Ref [22]. The waveguide lifetime of 836.8ns that the authors extract from their resonant-excitation measurements is much closer to the similar measurement presented in Ref [24] which reported 810ns lifetime for T centres in waveguides under resonant excitation. Could the difference in measurement technique (above-band vs resonance) account for the difference in lifetime? If so, should the modelling done in this paper use a 940ns bulk lifetime or the 836.8ns lifetime?
- Line 143 suggests that isotopically enriched SOI will minimize the T centre dephasing. Is the dephasing in question optical or spin dephasing?
- In Sect 3.1 of the SuppMat, the authors describe their method of using isolated T centre peaks to calibrate the T centre concentration in their samples. Were g_2 measurements performed on these isolated peaks to confirm they were single emitters? T centres can exist in a number orientations. Did the concentration estimates account for T centres in orientations that will couple poorly to the integrated devices?
- In Sect 5.5 of the SuppMat, the measured TX0-TX1 activation energy of 1.35meV differs quite dramatically from the expected value of 1.76meV. Is there an explanation for this difference? Perhaps strain from the integration into a device?

Authors' response to reviewers' report on manuscript NCOMMS-23-57828 entitled: "Cavity-coupled telecom atomic source in silicon."

Reviewer #1 (Remarks to the Author):

The article presents the embedment of Silicon T-centers in a photonic structure consisting of a nanobeam cavity, connected to a tapered waveguide and a grating to outcouple the photons into an optical fiber. On top of an impressive cleanroom work to fabricate the structure and generate the emitters, a methodical optical analysis has been performed, with a thorough estimation of the efficiency of the system, as well as complete characterization of one emitter. An impressive Purcell factor of 6 has been achieved, and all results have been compared to numerical modelling using JC Hamiltonian and Lindblad master equations.

The article is well structured, focused first on the fabrication and basic characterization of the sources, then on the estimation of the Purcell factor based on these measurements and on numerical modelling. While the article describes interesting results, it would further improve if the authors could clarify a certain number of points, see below:

The sample A presented in Figure 2c. does not seem to be used anymore afterward in the manuscript. If the only purpose of the sample A is to show that a higher fluence leads to a higher density of emitter, it would be helpful to mention it explicitly.

We thank the referee for the suggestion. Indeed, the purpose of including sample A was to show that the density of T centers can be controlled by the implantation fluence of carbon and hydrogen. We have modified the text as follows (in line 68 of the revised manuscript).

"The two samples shown in this work have an implantation fluence of $5 \times 10^{13} \text{ cm}^{-2}$ (sample A) and $7 \times 10^{12} \text{ cm}^{-2}$ (sample B), respectively, which result in different T center densities after the generation process."

Comparing these densities and the dimensions of the cavity, it could have been interesting to hint the reader on a "fabrication yield", that is, a probability to generate an emitter at the correct location (and maybe polarization) inside the photonic crystal.

We thank the referee for the suggestion. There are two aspects that are relevant here: (1) given the implantation fluence, the fabrication yield (i.e. probability) of generating T centers; 2) given the generated T center density and the cavity volume, how many T centers will be in the cavity.

For aspect (1), the estimated T center area density is $7.8 \times 10^6 \text{ cm}^{-2}$ and $2.5 \times 10^6 \text{ cm}^{-2}$ for sample A and sample B, respectively. When compared with their implantation fluences, we can obtain the fabrication yield to be 1.6×10^{-7} and 3.6×10^{-7} for sample A and B, respectively. This low yield has a contribution from the three-atom nature of the T center. Further optimization of our T center generation recipe is necessary to improve the yield.

For aspect (2), we first calculate the cavity volume of $0.0817 \mu\text{m}^3$, where the T center, if in it, can obtain appreciable enhancement (i.e. with simulated Purcell factor $P > 0.1 P_{\text{max}}$). We note that this volume is larger than the cavity mode volume. Using the estimated T center volume density, we then can expect around 0.082 and 0.025 T centers per cavity for sample A and sample B, respectively. These low average numbers of T centers per cavity qualitatively match with our experimental effort of measuring tens of cavities to find one with a T center inside.

In the above estimations, the dipole orientation or polarization is not considered. Further investigation is needed to reveal the nature of T center optical transition dipoles, and which T center orientations can have transition dipoles that couple well to the waveguide mode. Considering this, all the T center densities we estimated would be the lower bound since T centers with unfavorable orientations cannot be detected.

We have added the following sentences in the SI about the number of T centers in the cavity (starting line 150 of the revised SI),

“... Using a cavity volume where single T centers can obtain appreciable enhancement (i.e. with simulated Purcell factor $P > 0.1 P_{\text{max}}$), and the above T center densities, we estimate around 0.082 and 0.025 T centers per cavity for devices in sample A and sample B, respectively...”

Right after mentioning the densities, you suddenly discuss the emission efficiency of a single emitter into the waveguide mode? The link with the beginning of the paragraph is not straightforward. Also, you evaluate this η_{wg} for a single T-center at 46 GHz using the quantum efficiency of another T-center (the one that you use for the rest of the article at 11.3 GHz), although it seems that the first one is located inside the taper while the second one is located inside the nanobeam PC? Would the η_{wg} of these two emitters be identical despite their different position and eventually polarization?

We thank the referee for pointing out this potential source of confusion, and related questions.

We have added the following sentence in the paragraph to make the content read logically smoother (in line 77 of the revised manuscript),

“... To gauge the probability of an excited T center emitting a photon into the waveguide mode, we analyze a typical waveguide-coupled T center (at 46 GHz in sample B); ...”

Regarding quantum efficiency, we define it as $\eta_{QE} = (\gamma_{ZPL} + \gamma_{PSB})/\Gamma_0$, for T centers that are not coupled to a cavity. Thus, this formulation of quantum efficiency can be used for waveguide-coupled T centers.

Using the measurement results of the cavity-coupled T center at 11.3 GHz in sample B, we lower-bound the above-mentioned quantum efficiency, and subsequently use it to estimate the η_{wg} for the waveguide-coupled T center at 46 GHz in sample B. Here η_{wg} is the probability of a waveguide-coupled T center emitting a photon into the waveguide mode. Further details of the calculation can be found in Section 4 of the original Supplementary Information.

For cavity-coupled T centers, their emission probability to the cavity is characterized by the β factor. For the analyzed cavity-coupled T center at 11.3 GHz in sample B, we have $\beta = 85.5\%$ (see the last paragraph above the Discussion section in the main text). This β factor is the multiplication of the cavity-modified quantum efficiency and the Debye-Waller factor, which simplifies to $\beta = P_t/(P_t + 1)$, where P_t is the Purcell factor describing the fluorescence decay enhancement due to the cavity. After the T center fluorescence is emitted to the cavity, the probability of the photon being coupled out of the cavity to the taper waveguide is characterized by $\eta_{cav} = \kappa_{wg}/(\kappa_{wg} + \kappa_{sc})$, where κ_{wg} and κ_{sc} are the waveguide and scattering loss channels of the cavity, respectively. For the cavity that contains the single T center we discussed in the paper, we have $\eta_{cav} = 0.358$. The final probability of a cavity-coupled T center emits a photon into the waveguide mode will be $\beta \times \eta_{cav} = 0.855 \times 0.358 = 0.306$.

Indicating the detected count rates in pair with the counts per pulse could have been helpful for the readers, especially for such telecom applications where the actual count rate is of key importance.

We thank the referee for the comment. We mentioned these two pieces of information together in the original manuscript, as the following sentence shows,

“Under the pulsed excitation, the system can detect 0.01 ZPL photon per excitation, reaching an average photon count rate of 73.3 kHz, ...”

We calculate the average count rate by using the counts per pulse divided by the cavity-enhanced T center fluorescence lifetime when the cavity is resonant with the T center optical transition. This count rate is equivalent to the rate one can achieve for the cavity-coupled T center under CW laser excitations.

Is the g^2 mentioned in the abstract obtained for the count rate mentioned right after? Moreover, it could have been interesting to compare these g^2 and count rates not only to the T-center literature, but also to other deterministic telecom emitters from other materials (for instance based on GaN color centers: sciadv.aar3580 and Nanophotonics 12.8 (2023): 1405-1419.)

We thank the referee for pointing out the clarification and for the suggestion to add references for telecom emitters in GaN.

The $g^{(2)}(0)$ mentioned in the abstract was under a measurement with an input excitation power of 2.47 nW. The count rate of 73.3 kHz was calculated when the cavity-coupled T center reached saturation with the cavity being resonant, under an input excitation power of 17.01 nW. To clarify this and avoid confusion, we modified the relevant sentence in the abstract as follows,

“Efficient photon extraction enables the system to achieve an average ZPL photon outcoupling rate of 73.3 kHz under saturation, ...”

To include the quantum emitters in nitride, we have added the references into following sentences in the first paragraph when we discuss telecom emitters (in line 21 of the revised manuscript), and in the place where we compare $g^{(2)}(0)$ results (in line 87 of the revised manuscript), as shown below,

“Significant progress has been made towards utilizing atomic defects with telecom optical transitions, leading to the discovery of single vanadium ions in silicon carbide [7], defects in gallium nitride [8], and single erbium ions in yttrium orthosilicate...”

“This is the lowest $g^{(2)}(0)$ value ever observed for single T centers, and is comparable or better than other defect-based telecom emitters in solids [7–10, 20, 27, 37].”

In the JC Hamiltonian, it is not clear to me why the drive Hamiltonian is defined as function of the operators for the T-center 2-level system rather than of the cavity.

We thank the referee for this question. We had discussed this in Section 6.1 of the original Supplementary Information.

While solving the Lindblad equation, we found out that to calculate the atom population dynamics it is equivalent to use either the semiclassical drive $H_{semi} = \frac{\Omega}{2}(\sigma_+ + \sigma_-)$ or the coherent drive $H_{coh} = A(a^\dagger + a)$ in the weak coupling regime with $\kappa \gg g \gg \Gamma_0$ (i.e. where we are for the cavity-coupled T center). The results are the same as long as we use a large enough (i.e. larger than the mean intracavity photon number) Fock space in the coherent drive case. This can be understood that in such a weak coupling regime, the mean intracavity photon

number is mainly determined by the optical excitation and the cavity loss κ , and has a negligible influence from the coupling rate g . Thus the coherent drive is equivalent to the semiclassical drive with the mean intracavity photon number determined by the analytical formula as shown in Eq. S10, which is derived using the coherent drive Hamiltonian. Given the equivalence, we chose to use the semiclassical drive to save simulation time. In the coherent drive case, a much larger Hilbert space is needed, especially under high excitation powers, which makes the computation very time consuming.

In figure 4a, the blue fitting curve (no detuning) does not seem to catch an apparent longer decay time (time delays above 1.5 ns). Is this second decay rate real or only related to measurement uncertainty. If this decay is related only to the wg-coupled T-centers, and considering the large repetition rate that you have, could not you integrate longer to get rid of that visible second decay? How long did you integrate for these measurements?

We thank the referee for this observation and related questions. We believe that the second decay rate is likely caused by the waveguide-coupled (WG-coupled) T centers. Unfortunately, the cavity-coupled T center (at 11.3 GHz) is spectrally located at the inhomogeneous distribution center of WG-coupled T centers. The fluorescence emission from those un-enhanced WG-coupled T centers likely contributes at a longer time scale in the Purcell-enhanced fluorescence decay curve of the cavity-coupled T center (Fig. 4a).

Our full experimental sequence consists of $N = 2 \times 10^6$ excitation pulses followed by fluorescence collection windows (Fig. 2b). Each individual pulse sequence is $8 \mu\text{s}$ long with a $5.55 \mu\text{s}$ collection window. We log the photon arrival time after each excitation pulse and construct the fluorescence decay histogram as shown in Fig. 4a. Longer integration (e.g. by increasing repetition N) does not help to get rid of that 'second decay'. On the other hand, by lowering the excitation power, the fluorescence decay for the cavity-coupled T center with the cavity being resonant can be well described by a single exponential decay with a similar decay time constant, but without any sign of the 'second decay'. We believe that at lower excitation powers, the emission of the WG-coupled T centers decreases more than the cavity-coupled T center, which causes the 'second decay' to be much less pronounced. In Fig. 4a, we plotted a decay at a high excitation power (17.01 nW) aiming to show the count rate for the cavity-coupled T center under saturation.

Despite these few remarks, the article is very complete, the writing is clear, the structure is logic, and each characterization is conducted rigorously.

Reviewer #2 (Remarks to the Author):

The authors have provided a timely, interesting, and thorough study of Purcell-enhanced T centres integrated into photonic devices. Their paper provides impressive device performance for an early demonstration, a careful analysis of their measurements, and a very helpful and appreciated SuppMat section. I certainly recommend this excellent work for publication in Nature Communications.

Although none of these questions should be considered blockers to publication, I do have the following questions for the authors:

Line 119 of the main text: As written, it appears that the values of κ and Γ_0 were extracted from the fitting procedure, but these parameters were defined to have these values earlier in the paragraph. Were $\kappa/2\pi=5.22\text{GHz}$ and $\Gamma_0/2\pi=169.3\text{kHz}$ obtained from the global fit? Or were they fixed parameters in that procedure?

We thank the referee for pointing out this potential confusion. Indeed κ and Γ_0 are given parameters that we use in the global fitting. They are not obtained from the global fitting.

We have modified the relevant sentence to clear out the confusion (in line 121 of the revised manuscript).

“The global fitting of the experimental data based on the numerical calculations [35] given the known κ and Γ_0 , reveals the full cavity-QED parameter set $(g, \kappa, \Gamma_0) = 2\pi \times (42.4 \text{ MHz}, 5.22 \text{ GHz}, 169.3 \text{ kHz})$, an excited-state...”

On a related note, the bulk lifetime of 940ns used by the authors was measured with above-band excitation in Ref [22]. The waveguide lifetime of 836.8ns that the authors extract from their resonant-excitation measurements is much closer to the similar measurement presented in Ref [24] which reported 810ns lifetime for T centres in waveguides under resonant excitation. Could the difference in measurement technique (above-band vs resonance) account for the difference in lifetime? If so, should the modelling done in this paper use a 940ns bulk lifetime or the 836.8ns lifetime?

We thank the referee for the comments and related questions. For modeling the cavity induced Purcell enhancement of an emitter, one should compare its cavity-enhanced fluorescence emission rate with the unmodified, ‘free-space’ emission rate (*Nat. Photon.* 9, 427–435 (2015)). In the T center case, this ‘free-space’ emission rate would be the fluorescence decay rate when

the T center is in bulk silicon without any nanostructures. We thus use the 940 ns bulk lifetime in our modeling as the $1/\Gamma_0$.

When the T centers are in the waveguide, their emission dynamics may be modified by the changed local photonic density of states compared with the bulk case. Furthermore, other potential mechanisms, as proposed in *Opt. Express* 31, 15045-15057 (2023), such as free exciton capture time or superradiant enhancement may also modify the T center emission rate in the waveguide compared with the bulk case, which of course necessitates further investigations.

We acknowledge the fact that the 940 ns was measured under above-band excitations. Mixed results have been shown in the literature regarding whether different measurement methods (e.g. above-band vs resonant excitation) would reveal different fluorescence lifetimes for T centers. In the paper *Opt. Express* 31, 15045-15057 (2023), fluorescence lifetimes of 960 ± 10 ns and 810 ± 10 ns for the same T center ensemble were measured under above-band and resonant excitations, respectively. In a similar waveguide structure as ours, another measurement of the fluorescence lifetime of multiple single T centers under above-band excitations revealed an average lifetime of 838 ± 7 ns (*Nano Lett.* 24, 319–325 (2024)), which is consistent with our measured average lifetime for waveguide-coupled T centers under resonant excitations. Further investigations of T centers' fluorescence decay in bulk silicon under the resonant excitation are necessary to verify their bulk lifetime, which, however, falls outside the main scope of the current manuscript.

Line 143 suggests that isotopically enriched SOI will minimize the T centre dephasing. Is the dephasing in question optical or spin dephasing?

We thank the referee for this question. In the context, the dephasing in question is optical dephasing. In the enriched silicon, the ensemble T center inhomogeneous linewidth was shown to be 33 MHz (reference [23] in the revised manuscript), which puts an upper bound on the single T center linewidth in such an enriched silicon substrate. By incorporating single T centers in enriched SOI, we aim to significantly narrow down the single T center linewidth (i.e. to improve the optical coherence). Of course, the enriched SOI can also help on the T center spin coherence by minimizing the density of ^{29}Si isotopes, which have nonzero nuclear spins.

In Sect 3.1 of the SuppMat, the authors describe their method of using isolated T centre peaks to calibrate the T centre concentration in their samples. Were g2 measurements performed on these isolated peaks to confirm they were single emitters? T centres can exist in a number orientations. Did the concentration estimates account for T centres in orientations that will couple poorly to the integrated devices?

We thank the referee for the comments and related questions. For those isolated peaks that are not coupled to the cavity, we performed $g^{(2)}$ measurements without any background correction. A typical $g^{(2)}(0)$ will go below 0.5, yet with a large error bar due to the low SNR for those peaks. Considering the narrow linewidth (~ 2 GHz), as well as the saturation behavior (see Section 4 in the Supplementary Information), we strongly believe that those isolated peaks originated from single T centers. This single emitter nature for waveguide-coupled single T centers, however, is not as decisive as the cavity-coupled T center we showed in the main text.

The concentration estimates we have in the manuscript did not consider the T center orientations. It has been shown that T centers can have 12 structural orientations in the silicon lattice (*PRX Quantum* 1, 020301246 (2020)). However, further investigation is needed to reveal the nature of T center transition dipoles, and how many T center orientations can have transition dipoles that couple well to the waveguide mode. Considering this, we did not take into account the T center orientations during the density estimation, and all the estimated densities would be the lower bound of the actual T center density.

In Sect 5.5 of the SuppMat, the measured TX₀-TX₁ activation energy of 1.35meV differs quite dramatically from the expected value of 1.76meV. Is there an explanation for this difference? Perhaps strain from the integration into a device?

We thank the referee for the observation and related questions. The strain due to the device integration can be a possible reason for this discrepancy. To verify this, we are trying to measure the TX₀-TX₁ splitting via PLE spectroscopy for ensemble T centers in bulk silicon, unpatterned SOI, and patterned SOI, to confirm whether the splitting is affected by the strain from the device patterning. These further investigations, however, fall outside the main scope of the current manuscript. We will try to include them in our future works. Another plausible cause of the discrepancy can come from the measurement uncertainty. Due to technical limitations, we can't reach temperatures below 3.4 K and thus miss important points that may affect the fitting results in Fig. S15.

REVIEWERS' COMMENTS

Reviewer #1 (Remarks to the Author):

Authors have revised according to the previous comments and I recommend its publication now.

Reviewer #2 (Remarks to the Author):

I would like to thank the authors for their thorough and satisfactory responses to my review. At this point, I approve this manuscript for publication with no further revisions.